# Hydroxytyrosol and Brain Tumors: Mechanisms of Action and Therapeutic Potential

**DOI:** 10.3390/cimb47080667

**Published:** 2025-08-18

**Authors:** Cristina Cueto-Ureña, María Jesús Ramírez-Expósito, María Pilar Carrera-González, José Manuel Martínez-Martos

**Affiliations:** Experimental and Clinical Physiopathology Research Group CTS-1039, Department of Health Sciences, School of Health Sciences, University of Jaén, E-23071 Jaén, Spain; ccueto@ujaen.es (C.C.-U.); mramirez@ujaen.es (M.J.R.-E.); pcarrera@ujaen.es (M.P.C.-G.)

**Keywords:** hydroxytyrosol, glioblastoma, oxidative stress, blood–brain barrier, natural antioxidants, central nervous system tumors

## Abstract

Central nervous system (CNS) tumors, especially gliomas and IDH-wildtype glioblastoma, present high aggressiveness, low response to current treatments and limited survival. Several biological processes such as oxidative stress, inflammation, apoptosis, and autophagy are involved in their development. Hydroxytyrosol (HTX), a phenolic compound present in olives, has shown relevant effects on these processes in experimental models. This review analyzes its chemical characteristics, bioavailability, and ability to cross the blood–brain barrier, as well as its mechanisms of action. Despite its rapid metabolism, HTX can reach the brain in small but functional amounts, and various formulation methods can enhance its delivery to nervous tissue. HTX acts on cellular pathways such as Nrf2, NF-κB, JAK/STAT, PI3K/Akt and SIRT1, regulating redox balance, inflammation, programmed cell death, and autophagy. It can also influence gene expression through epigenetic mechanisms. In cell models, it has shown inhibitory effects on tumor growth and activation of apoptosis, without affecting non-tumor cells. These results support its possible usefulness as an adjunct in the treatment of brain tumors, although further studies in animal and human models are required.

## 1. Introduction

Tumors affecting the central nervous system (CNS) constitute a heterogeneous group of neoplasms with complex biology and for which effective treatments remain elusive [1,2]. These tumors can originate from diverse cells and structures in the brain and spinal cord, showing wide variability in histology, clinical behavior, and prognosis [1,2]. Among primary CNS tumors, gliomas are the most frequent, especially those that develop in the brain parenchyma [1,2]. According to recent data, gliomas account for approximately 23% of all primary CNS tumors, and constitute 51.5% of CNS malignancies [3,4,5,6,7].

The WHO classification of 2021 introduced fundamental changes in the diagnosis of glioblastoma, establishing molecular criteria that allow the diagnosis of “molecular glioblastoma” even in the absence of classical histological features [1,8]. According to this new classification, IDH-wildtype glioblastoma can be diagnosed when at least one of three specific genetic criteria is identified: TERT promoter mutation, EGFR amplification, or combined +7/−10 chromosomal alterations [9,10,11]. These molecular criteria are particularly relevant for tumors that histologically appear to be lower grade but behaviorally resemble traditional glioblastoma [10,11]. Implementation of these criteria has resulted in the reclassification of approximately 30% of low-grade gliomas as high-grade tumors, significantly improving prognostic accuracy [10]. These tumors are characterized by their rapid growth, high infiltration capacity in the surrounding brain tissue that makes complete surgical resection difficult and favors recurrence, and the resistance that many brain tumors present to chemotherapy and standard radiotherapy [2], leading to a very unfavorable prognoses [12]. The five-year relative survival in Spain for patients with glioblastoma is around 6–7% [13]. A French multicenter study involving 1438 patients with high-grade gliomas treated with standard chemoradiotherapy reported a median overall survival of 20.4 months, with survival rates of 78.5% at one year, 40.3% at two years and 11.8% at five years [14]. A targeted analysis of 976 patients with IDH-wildtype glioblastoma confirmed a median survival of 11.2 months (12.2 months for patients treated after 2005), with 17.6% of patients achieving survival at two years and 2.2% at five years [15]. These updated data reflect modest but significant improvements in prognosis, possibly related to advances in surgical techniques, radiotherapy protocols and multidisciplinary management [15,16,17,18,19,20,21]. In any case, these studies reflect the high lethality of the disease and the limitations of current treatments, so it is necessary to advance in the development of more effective therapies. Conventional pharmacological therapies remain a mainstay in the treatment of CNS tumors, but have important limitations, such as the multiple mechanisms of resistance to temozolamide, the most widely used antitumor drug [22,23,24]. Frequently, these treatments are associated with adverse side effects, such as hematological, gastrointestinal and neurological toxicity, which can significantly affect patients’ quality of life [22,25]. Moreover, in many cases, the primary goal of these therapies is to alleviate clinical symptoms and temporarily slow disease progression, rather than to achieve complete remission or sustained long-term control [22,25], which remains a considerable challenge today [25]. In any case, the glioma therapeutic landscape has seen significant advances with the approval of vorasidenib by the FDA in August 2024 for grade 2 gliomas with IDH1/IDH2 mutations [26]. This dual IDH1/IDH2 inhibitor demonstrated in the phase III INDIGO trial a median progression-free survival of 27.7 months compared to 11.1 months for the placebo, representing more than a two-fold improvement [26,27]. Vorasidenib is the first systemic treatment approved specifically for this patient population, marking a milestone after more than 20 years without new therapeutic options for low-grade gliomas [28].

While gliomas represent the largest group of primary brain tumors, other types of central nervous system neoplasms have distinctive characteristics in terms of incidence, biological behavior, and prognosis. The 2021 WHO classification has introduced significant changes in the categorization of these tumors, establishing specific molecular criteria for several of these neoplastic types [1,29]. Thus, medulloblastoma is the most common embryonic tumor in childhood, accounting for approximately 20% of all pediatric brain tumors. Four main molecular subgroups are distinguished: WNT-activated medulloblastoma, SHH-activated medulloblastoma with wild-type TP53, SHH-activated medulloblastoma with mutated TP53, and non-WNT/non-SHH medulloblastoma [30]. Each subgroup has distinctive clinical-pathological characteristics and different prognoses. WNT-activated medulloblastoma is associated with a favorable prognosis, with 5-year survival rates exceeding 90%. In contrast, non-WNT/non-SHH tumors show a greater propensity for metastatic spread, with metastatic disease rates at diagnosis of 30–45% and 5-year survival rates of 50–75%. The overall 5-year survival rate for medulloblastoma is approximately 70–80% when the disease has not spread, falling to 60% in cases of metastatic disease. Multimodal treatment includes maximum safe surgical resection, chemotherapy, and craniospinal radiotherapy, with the extent of surgical resection being a significant prognostic factor [1,30,31].

Ependymomas have undergone substantial reclassification in the 2021 WHO classification, with the definition of eight specific types based on anatomical location and molecular characteristics [1,29]. This new classification includes supratentorial ZFTA-fusion-positive ependymoma, supratentorial YAP1-fusion-positive ependymoma, posterior fossa ependymomas groups PFA and PFB, spinal ependymoma, and MYCN-amplified spinal ependymoma. Survival rates vary significantly depending on molecular subtype and location, with an overall 5-year survival rate of 85%. Spinal ependymomas have the best prognosis with a 10-year survival rate of 100%, while supratentorial ependymomas have a 10-year survival rate of 20%. In the pediatric population, 10-year survival is approximately 50–70%. The main treatment includes surgical resection followed by radiotherapy, with chemotherapy reserved for specific cases such as very young children or recurrent disease [1,29,32,33].

Craniopharyngiomas have been reclassified in the WHO 2021 as two distinct entities: adamantinomatous craniopharyngioma and papillary craniopharyngioma, reflecting their different clinical-demographic and molecular characteristics [1,29]. Despite being histologically benign tumors (WHO grade 1), they have a high propensity for recurrence and significant morbidity due to their sellar/suprasellar location. The 3-, 5-, and 10-year survival rates are 89.1%, 86.2%, and 83%, respectively. In the pediatric population, the 5-, 10-, and 15-year survival rates are 100%, 98.3%, and 94.6%, respectively. Factors associated with better survival include younger age, smaller tumor size, and subtotal resection combined with radiotherapy. Current management tends towards conservative strategies combining limited surgery with adjuvant radiotherapy to minimize hypothalamic morbidity [34,35,36,37].

CNS embryonal tumors other than medulloblastoma include atypical teratoid/rhabdoid tumor (ATRT), embryonal tumor with multilayer rosettes (ETMR), and CNS neuroblastoma with FOXR2 activation [1]. ATRT, characterized by the loss of SMARCB1, has a very poor prognosis with 5-year survival rates of 47.8% in children, 41.5% in adolescents, and 24.6% in adults. ETMR tumors show an overall 5-year survival rate of approximately 25% in retrospective series, reaching up to 66% in patients treated with intensified protocols. Survival is particularly poor in infants and patients with metastatic disease [38].

Diffuse midline gliomas (formerly DIPG) represent highly aggressive brain stem tumors with an extremely poor prognosis. Median survival is 8–11 months, with 2-year survival rates of approximately 10% and 5-year survival rates of 2%. Factors associated with prolonged survival include age younger than 3 years or older than 10 years, shorter duration of symptoms, smaller tumor size, and presence of HIST1H3B mutation [39,40,41].

Primary CNS lymphomas are predominantly diffuse large B-cell lymphomas with an overall survival of 12–18 months without treatment. With high-dose methotrexate-based therapy, 2-year survival rates reach 42–80.8%. However, the prognosis for refractory disease or relapse is very poor, with a median survival of 2 months without treatment [42,43].

There is, therefore, considerable biological and clinical heterogeneity among non-glial brain tumors.

Tumor development in the CNS is associated with multiple biological phenomena, among which the role of oxidative stress stands out. Oxidative stress is defined as an imbalance between the production of reactive oxygen species and the antioxidant capacity of the organism, and has been implicated in carcinogenesis due to its capacity to induce damage to DNA, proteins, and cellular lipids [2,44,45]. The transformation of a normal cell into a malignant cell in the CNS is a multistage process involving molecular and cellular alterations, in which oxidative stress can act as a triggering or facilitating factor [2,44,45]. Additional processes such as neuroinflammation, apoptosis, autophagy or stress response are also involved. The exploration of natural bioactive compounds with neuroprotective, antioxidant and anti-inflammatory properties, such as hydroxytyrosol (HTX), and the study of their ability to cross the blood–brain barrier (BBB) represent a promising avenue in the approach to CNS tumors [46,47].

## 2. Chemical Characteristics of HTX

HTX is a natural phenolic compound that has shown increasing therapeutic potential [9,48,49]. Chemically, it corresponds to the molecule 2-(3,4-dihydroxyphenyl)ethanol [46]. Its structure, which includes a catecholic moiety, is key to its potent antioxidant activity [44,50]. This feature allows it to participate in redox cycles and oxidize to its corresponding catecholic quinone [44,50]. HTX is considered one of the most potent natural antioxidants [9,44,46,47,48,49,50,51,52,53,54,55], capable of neutralizing reactive oxygen and nitrogen species, protecting neuronal cells from oxidative damage [44,48,50,52,53,54]. It may also modulate endogenous antioxidant systems, as well as multiple intracellular signaling systems, to contribute to its neuroprotective effect [52,53].

### 2.1. Synthesis and Metabolism of HTX

The natural biosynthesis of HTX is closely linked to olive (*Olea europaea* L.), since, although it is present in the plant, most of its content in olives and derivatives comes from the hydrolysis of oleuropein (OLE), a secoiridoid especially abundant in the leaves, which contains a structural subunit known as 4-(2-dihydroxyethyl)benzene-1,2-diol, corresponding to HTX [44,48,50,56,57]. OLE is hydrolyzed by endogenous enzymes such as β-glucosidases, releasing glucose, oleuropein aglycone, elenolic acid, and HTX [58]. During olive ripening and oil extraction processes [59], both virgin (VOO) and extra-virgin (EVOO), OLE undergoes several transformations to originate HTX [46,52,56,60]. Its content in EVOO varies between 3.0 and 25.6 mg/kg, depending on factors such as olive variety, climate, fruit maturity, harvest and production process [50,56,61]. Refining removes these compounds, so they are found almost exclusively in VOO and especially in EVOO [44,62]. Olive leaves and by-products from olive processing, such as olive pomace and vegetation water, are alternative sources of HTX (Figure 1), and methods exist for their large-scale recovery, such as Hidrox^®^ extract obtained by mild acid hydrolysis [54,56].

In addition, the acid hydrolysis of OLE that occurs in the stomach after ingestion releases glucose and an aglycone from which HTX can be derived [44,63]. In humans, HTX has been reported to constitute between 96% and 99% of the phenolic metabolites of OLE detected in plasma, underscoring OLE’s role as a major source of HTX [48]. In addition, intestinal microflora can degrade OLE in the colon using enzymes such as β-glucosidases, further contributing to the generation of HTX, which can also exert biological effects locally [64].

On the other hand, HTX can be produced endogenously as a byproduct of the oxidative metabolism of dopamine. Monoamine oxidase (MAO) catalyzes the formation of DOPAL (3,4-dihydroxyphenylacetaldehyde), a neurotoxic metabolite that can be reduced to HTX by the action of enzymes such as aldehyde reductase. This pathway could represent a physiological detoxification mechanism, as HTX can inhibit both enzymatic and spontaneous oxidation of endogenous dopamine [65].

HTX undergoes extensive hepatic and intestinal metabolism, resulting in multiple conjugated metabolites, which are widely distributed in tissues such as the liver, kidney, lungs, heart, skin, and spleen [66]. Studies have shown that they can accumulate in mitochondria and cross the BBB in small amounts. Some metabolites, such as caffeic acid sulfate, reach relatively high concentrations in the brain, suggesting the existence of selective transport mechanisms [67,68].

The biotransformation of HTX occurs in two main phases. In phase I, hydrolysis and/or hydroxylation reactions occur, catalyzed by enzymes such as cytochrome P450. Hydroxylated and hydrated derivatives can be generated in the liver and intestine. In phase II, conjugation reactions predominate, facilitating solubility and excretion. Several processes occur [66,68,69]. On the one hand, sulfation, catalyzed by sulfotransferases (SULTs), predominates at high doses of HTX. Sulfated conjugates account for approximately 73% of all circulating metabolites. Secondly, glucuronidation is mediated by UDP-glucuronosyltransferases (UGTs), which are dominant at low doses. Glucuronides account for only 2–4% of total metabolites, with AUC ratios of 0.15–0.25 relative to HTX. Thirdly, methylation occurs by catechol-O-methyltransferases (COMT), which produces metabolites such as homovanillic acid (HVA) and 3,4-dihydroxyphenylacetic acid (DOPAC). These metabolites reach maximum plasma concentrations approximately 30 min after ingestion. Finally, acetylation takes place via acetyl transferases (ACTs), which generate HTX acetate (HTX-ac). Among the metabolites identified are 3,4-dihydroxyphenylacetic acid, 3,4-dihydroxyphenylacetaldehyde sulfate, homovanillyl alcohol, homovanillic acid, and various glucuronide conjugates [66,68,69].

### 2.2. Bioavailability of Hydroxytyrosol

The bioavailability of HTX is a key factor for its therapeutic potential. Although phenolic compounds usually have a low bioavailability, animal and human studies have shown that HTX is rapidly absorbed in the small intestine and colon, reaching maximum plasma concentrations (Cmax) between 30 and 62 min after oral administration. Due to its low molecular weight, it has a large distribution to the brain and other organs [54], where it can exert biological effects in a dose-dependent manner [44,50,54,55]. Additionally, the bioavailability of HTX after oral administration depends on the delivery matrix [66]. In fact, when administered in olive oil, bioavailability is 99%, while in aqueous solution, bioavailability drops to 75%.

As HTX is rapidly metabolized after ingestion, its conjugated metabolites are the ones mainly detected [50,56,70,71]. HTX undergoes phase II metabolism, mainly through the sulfation and glucuronidation pathways [68,69]. The sulfation pathway predominates, with exposure between 7 and 9 times higher than that of the original compound, while the glucuronidation pathway has exposure <0.25 times higher compared to the original compound. Only ~11% of the total circulating compounds appear as free HTX. The main conjugated metabolites are two sulfate conjugates and two glucuronide conjugates. Certainly, these multiple metabolites retain biological activity at the cellular level [44], although with varying potencies compared to the original compound. HTX-3-O-sulfate and HTX-4-O-sulfate maintain significant biological activity (between 60 and 80% of the original), while HTX-3-O-glucuronide and HTX-4-O-glucuronide retain moderate biological activity (between 40 and 60% of the original), especially in relation to their antioxidant properties [68,69]. Some secondary metabolites such as 3,4-dihydroxyphenylacetic acid (DOPAC), homovanillic acid (HVA), and homovanillic alcohol still show antioxidant and cytoprotective activity, although lower than that of the original compound (between 20 and 40% of the original) [68,69]. Therefore, the metabolism of hydroxytyrosol implies that its biological activity cannot be attributed solely to the original compound, and that it is probably the sulfate conjugates, which represent 73% of the circulating metabolites, that contribute most to the observed benefits [50,56,70,71].

Bioavailability may also vary according to gender and the presence of other polyphenols in EVOO, suggesting possible synergistic interactions that enhance their effects [56,72,73]. One study found that certain phenolic metabolites, such as 3-O caffeic acid sulfate, showed higher peak concentration values and area under the curve in brain tissue compared to non-brain areas, suggesting a possible enhanced transport mechanism through the BBB for certain metabolites [61,74].

Among the strategies explored to improve HTX absorption and bioavailability is the use of ester derivatives, such as hydroxytyrosyl oleate (HTX-OL), encapsulated in solid lipid nanoparticles, which has been shown to improve cellular absorption, stabilize the molecule, and allow controlled release [75]. Other derivatives, such as HTX octanoate and decanoate, have shown improved oral bioavailability by promoting a progressive release during digestion [75]. Liposomal encapsulation protects HTX, increasing its biological activity and therapeutic potential, while organogelated emulsions with soybean oil have shown excellent gastrointestinal absorption and physicochemical stability [75].

### 2.3. Hydroxytyrosol and the Blood–Brain Barrier

The main obstacle for the treatment of CNS tumors is the BBB [2,76]. The BBB is a highly specialized structure that separates the vascular compartment from the central nervous system, tightly regulating the exchange of substances between the blood and the brain parenchyma [77,78]. Structurally, the BBB consists mainly of endothelial cells of the cerebral capillaries, which are characterized by the presence of tight junctions that restrict the paracellular passage of molecules and by a notable absence of fenestrations and low pinocytic activity. These cells are supported by a basement membrane and surrounded by pericytes, which modulate both the structural stability and function of the endothelium. Perivascular feet of astrocytes externally envelop capillaries and release factors that induce and maintain the barrier properties of the endothelium, although they are not a direct part of the physical barrier [77,78]. Microglia and neurons, although not integral to the BBB structure, are involved in signaling and functional modulation within the neurovascular unit, a concept that highlights the dynamic interaction between these elements. In addition, recent studies have evidenced regional heterogeneity in BBB composition and function, which implies differences in susceptibility to pathologies and drug distribution according to the brain region considered [77,79].

The BBB performs regulatory and protective functions through mechanisms that ensure strict selectivity in the transport of substances. It only allows the passage of essential nutrients, such as glucose and amino acids, through specific transporters, while restricting the entry of toxins, pathogens and most hydrophilic drugs [77]. This control is achieved by protein-mediated transport systems, such as GLUT1 for glucose, and efflux pumps, such as P-glycoprotein, which actively expel potentially harmful compounds into the blood. Passive diffusion is limited to small, lipophilic molecules. In addition, the BBB maintains ionic and osmotic homeostasis of the brain microenvironment, regulates pH and facilitates the elimination of metabolic waste products. Its role in the immune response is relevant, as it limits the entry of immune cells and proinflammatory molecules, contributing to the immunoprotection of the central nervous system [77,78,79]. Finally, the BBB participates in neurovascular communication, integrating neuronal, vascular and glial signals to adjust both blood flow and permeability according to brain metabolic needs [2,79,80,81]. However, it also significantly limits drug delivery to the brain, preventing most therapeutic agents from reaching effective concentrations at the tumor site [76,77,78,79,80,81,82], so overcoming the BBB is a central goal in new drug development [46,47,82]. It is estimated that a penetration through the BBB of just 1–5% of the administered dose or plasma concentration may be sufficient for a drug to exert its action in the brain [47,82]. Therefore, current research is focused both on identifying compounds with an inherent ability to cross the BBB and on developing strategies, such as nanotechnology or chemical modifications, that enhance their transport [47,82]. In any case, it has been shown that HTX can cross the BBB and be present in brain tissue [44,47,61,62,83]. In fact, HTX absorption is rapid, and several factors determine HTX penetration into the CNS, including its low molecular weight, passive transport, active transport mediated by monocarboxylic acid and sodium-dependent transporters, and the lipophilicity or polarity of its metabolites [56,61] (Figure 2). In any case, although in vitro and in silico studies suggest that simple diffusion, influenced by lipophilicity, may play a role in BBB crosstalk, in vivo studies indicate that the mechanism remains unclear and alternative processes such as the use of specific transporters or paracellular and vesicular transport cannot be ruled out [61].

As approximately 98% of small-molecule drugs and most large-molecule neurotherapeutic agents are excluded from the brain by the BBB, efforts are underway to optimize effective delivery across the BBB for the achievement of therapeutic concentrations in brain tissue [84,85].

Various phenolic compounds and their derivatives have been evaluated in in vitro models of BBB permeability, such as the PAMPA assay. In this context, HTX and its ester, HTX acetate (HTA), have shown promising effective permeabilities (Pe) [46]. HTA, in particular, presented a Pe of 2.54 ± 0.2 (10^−6^ cm s^−1^), twice that of verapamil, known for its high permeability, while HTX had a Pe of 1.46 ± 0.04 (10^−6^ cm s^−1^), and both values higher than those of theophylline (low permeability) and L-dopa (a drug that crosses the BBB) [46]. These results suggest that esterification of hydroxyl groups, as in HTA, could be an effective strategy to increase the permeability of these compounds [46].

The conjugated antioxidant TPP-HT, which combines HTX with triphenylphosphine (TPP), a group with mitochondrial affinity, has also been developed [47]. According to Panara et al., this compound, when administered orally to mice, was shown to penetrate the BBB and was quantified in the cerebellum, reaching approximately 5% of the total administered (11.45 ± 0.65 ng g^−1^), which is considered adequate for a brain-targeted drug [47]. TPP-based compounds can achieve a 50- to 70-fold higher concentration in the mitochondrial matrix, suggesting that, although the amounts detected in brain tissue may be low, specific accumulation in mitochondria could result in a significant concentration at the subcellular level [47]. This TPP-HTX derivative, which accumulates in the cerebellum at a rate more than twice that in the liver, is considered promising for in vivo experiments aimed at modulating brain mitochondria. These findings further underscore the potential of natural product-derived compounds, such as HTX and its modifications, to address the challenge of drug access to the brain [46,47].

Other in vivo studies have detected that HTX or its metabolites appear in the brain of rats after intravenous or intraperitoneal administration, after 15 min of administration. Sulfated metabolites are the most commonly detected in the brain [61]. Similarly, administration of OLE in neonatal rats allowed detection of small amounts of OLE in the brain, and maternal supplementation with OLE resulted in accumulation of its metabolite, HTX, in the offspring’s brain [61]. A recent study with the HTHB derivative, orally administered to mice, showed that this compound crosses the BBB and was quantified in the cerebellum [47,86]. In rat models of induced Alzheimer’s disease, HTX was detected in the frontal cortex and hippocampus after injection [83].

However, it should be noted that, although HTX is rapidly absorbed and reaches peak plasma concentrations within minutes, its systemic bioavailability is limited by phase I and II metabolism in the intestine and liver [53,62,70,71], so HTX is excreted mainly in urine and has a short plasma half-life (approximately 2.5 h) and a mean lifetime of around 4 h [61]. To overcome the low systemic bioavailability, alternative routes of administration (intravenous, intraperitoneal) as well as chemical modifications of HTX have been explored. Nanotechnological strategies to improve the delivery of therapeutic compounds to the brain have undergone significant advances, which is particularly relevant for optimizing HTX bioavailability [87]. Solid lipid nanoparticle (SLN) and nanostructured lipid carrier (NLC) systems have demonstrated superior ability to cross the blood–brain barrier via receptor-mediated transcytosis mechanisms [88,89,90,91,92].

The development of derivatives such as hydroxytyrosyl oleate (HTX-OL) encapsulated in lipid nanoparticles has shown improved cellular uptake, molecular stabilization and controlled release [93,94]. Carbon-polymer hybrid systems have shown ability to reach the hippocampus one hour after administration, while lecithin-chitosan formulations have increased brain bioavailability [95]. Conjugation with mitochondria-targeted groups, such as the TPP-HTX derivative, allows 50–70-fold higher specific accumulation in the mitochondrial matrix, optimizing subcellular effects [47,84]. It has been shown that HTX-3 nanoparticles, a derivative of HTX conjugated to caffeic acid skeletons, can enhance BBB penetration, accelerate metabolism, and prolong brain retention in murine models [96]. These nanoscale delivery systems take advantage of physicochemical properties and biocompatibility to enhance delivery to the CNS [84]. Although not directly tested with HTX, loading urolithins into milk exosomes enhanced their delivery to the brain, illustrating the potential of exosomes as biological delivery vehicles for bioactive compounds to the brain [61]. Finally, natural formulations such as alpechin extracts may show superior effects to pure HTX, although further research on the mechanisms involved is required [61].

Therefore, from a pharmacokinetic point of view, it can be assured that HTX can cross the BBB [56], and although the percentage that crosses it is low, compounds targeting mitochondria can accumulate significantly in these organelles [47]. It has also been shown that in those situations in which BBB permeability is increased, as occurs in some animal models of depression, HTX concentrations reached in nervous tissue are higher than in healthy controls [61]. A possible interaction with the dopaminergic pathway has also been proposed [61].

## 3. Mechanisms of Action of HTX

### 3.1. Antioxidant Properties and Modulation of Oxidative Stress

Tumorigenesis in the CNS is characterized, among other factors, by a strong redox imbalance that not only contributes to its progression, but also to therapeutic resistance [44,45,97]. This occurs because the CNS is particularly sensitive to free radical damage, and tumor development involves multiple cellular and molecular events that appear to be triggered by these [45].

Increased oxidative stress in tumor cells is manifested by increased lipid peroxidation, the presence of carbonyl and diene-conjugated groups (protein oxidation) and decreased total antioxidant capacity (TAC) in tumor tissue [45,62,98]. Similarly, decreased levels of glutathione (GSH), a key non-enzymatic antioxidant, are observed in both brain tumor tissue and plasma of animals with induced tumors [45,62,98]. Antioxidant enzymes such as superoxide dismutase (SOD) and catalase (CAT) may also exhibit reduced activity in tumor tissue [45,62,98]. However, it is not only the level of individual antioxidant systems, but the imbalance in the activity and proper expression of antioxidant enzymes that seems most important in preventing or promoting tumor growth [45,62,98]. Cancer cells operate continuously under oxidative stress due to high intracellular generation of free radicals, promoted by increased metabolic activity and/or mitochondrial malfunction [45,62,98]. This oxidative damage is perpetuated by this reduced response of antioxidant defense systems, which facilitates further tumor development [45,62,98]. In this situation, HTX would protect healthy cells from oxidative stress-induced damage and death [45,50,73] and would act negatively on tumor cells.

In an experimental model of N-nitroso-N-ethylurea (ENU)-induced glioma in rats, oral administration of HTX, both alone and in combination with OLE, has shown limited beneficial effects on brain tumorigenesis. These effects correlate with the modulation of components of the endogenous redox system, such as increased antioxidant enzyme activity and reduced oxidative damage in tumor tissue. It is relevant to note that the magnitude of these effects is highly dependent on the sex of the animals, suggesting an interaction with the hormonal environment and the tumor microenvironment [44,62,97,98,99,100,101].

In a rat model of subcutaneous C6 glioma, HTX treatment maintained non-enzymatic antioxidant defense systems at levels similar to healthy animals and enhanced enzymatic systems, which is associated with less tumor progression [98,102]. Furthermore, these studies have evidenced that HTX can modulate the activity of key antioxidant enzymes such as SOD, CAT and glutathione peroxidase (GPX), as well as increase reduced GSH levels, contributing to a less favorable environment for tumor growth [98,102]. They also observed that oral administration of HTX caused a marked inhibition of tumor growth, an effect accompanied by a reduction in systemic inflammatory markers such as IL-6 and TNFα, as well as in the activity of peptidases related to the brain renin–angiotensin system (bRAS). Specifically, HTX modulated aminopeptidase enzymes that favor the production of antitumor peptides (ang1–7) against proinflammatory angiotensins [100,101,103]. Thus, HTX could induce a less favorable environment for the tumor, combining systemic anti-inflammatory effects with local modifications of the tumor microenvironment, such as reduced angiogenesis and improved perfusion [100,101,103].

However, the relationship between antioxidant and antitumor effects is complex. Although oxidative stress favors tumor progression, cancer cells with high levels of endogenous antioxidants may show greater resistance to therapies that rely on ROS to induce apoptosis [45,62,98,100,101,103]. In fact, administration of OLE in the ENU-induced rat glioma model significantly increased tumor volume in male animals, which could be explained by inhibition of the antitumor immune response, possible agonist effect on growth factors or reduction in ROS-mediated cell death due to its potent antioxidant activity [45,62,98,100,101,103]. This suggests that the delicate balance between pro-oxidants and antioxidants is crucial, and that the antitumor effect of HTX may involve mechanisms in addition to antioxidant action, possibly related to the modulation of intracellular signaling pathways influenced by hormonal status, especially in hormone-dependent tumors [45,62,98,100,101,103]. It also indicates that the effects of polyphenols are dependent on tumor type, experimental model, route of administration and sex of the animals.

HTX has also been shown to decrease oxidative damage in various cellular models by neutralizing pro-oxidants and directly scavenging reactive oxygen species (ROS) and reactive nitrogen species (RNS) [51,73]. In this way, lipid peroxidation, protein oxidation and DNA damage are prevented [73,104]. Thus, it has been demonstrated in vitro that HTX can protect neuroblastoma SH-SY5Y cells from oxidative damage induced by H_2_O_2_ and glutamate, as well as from toxic stress caused by S100A9 amyloid aggregates, improving mitochondrial functionality and reducing ROS production [50]. Furthermore, HTX has been effective in protecting cells against the neurotoxin 6-hydroxydopamine (6-OHDA), relevant in pathologies such as Parkinson’s disease, through antioxidant mechanisms [46]. HTX also alleviates oxidative stress and neuroinflammation, and enhances neurotrophic signaling of hippocampal neurons, resulting in an improvement in stress-induced depressive behaviors in murine models [105,106].

#### Signaling Pathways

In addition to direct free radical scavenging, HTX modulates key pathways of oxidative stress and endogenous antioxidant defense systems [73]. One of the most studied mechanisms is the activation of the Nrf2/ARE (nuclear factor erythroid 2-related factor 2/antioxidant response element) pathway [49,53,107]. Nrf2 is a critical transcription factor for protection against oxidative stress [107]. When activated by compounds such as HTX, Nrf2 accumulates and translocates to the nucleus, where it binds to ARE at gene promoters, inducing transcription of multiple antioxidant and phase II detoxification enzymes [50,54,96,104,107]. Among these enzymes are heme oxygenase-1 (HO-1), NAD(P)H:quinone oxidoreductase 1 (NQO1), thioredoxin (Trx) and thioredoxin reductase, γ-glutamylcysteine synthetase, glutathione S-transferase (GST), GPX, CAT and glutamate cysteine ligase (GCLC), with the latter involved in glutathione synthesis [9,50,52,96,104,107,108]. HTX derivatives, such as HTX butyrate (HTX-B), also activate Nrf2 by inhibiting the Keap1 protein, allowing Nrf2 stabilization and nuclear translocation [86,109]. It has also been described that the HTX-rich olive extract Hidrox^®^ activates Nrf2 and, for example, has been shown to counteract cyclophosphamide-induced male infertility in a murine model [53,107]. Overall, HTX contributes to mitigating oxidative stress and maintaining redox balance by modulating these enzyme systems, which is critical for their protective and potentially antitumor effects [45,62] (Figure 3).

Other signaling pathways, such as those mediated by PI3K/Akt and ERK1/2, involved in the activation of the Nrf2/HO-1 axis, are also enhanced in the presence of HTX, which reinforces the cellular antioxidant response mediated by this compound [73,110]. These pathways are also involved in neuroinflammation processes, and are therefore, also affected by HTX.

### 3.2. Anti-Inflammatory Properties

Neuroinflammation is a key brain process, closely linked to the activation of microglia cells, the resident immune cells of the brain [75,86].

HTX and other phenolic compounds have also been shown to differentially modulate multiple signaling pathways involved in neuroinflammatory processes. These pathways include NF-κB [50] and the JAK/STAT pathway [56]. Likewise, and as indicated above, HTX decreases LPS- and α-synuclein-induced microglial activation in vitro, suggesting a potential additional neuroprotective effect [64]. However, although HTX had the ability to reduce LPS-induced NF-κB translocation in microglia, it had no effect on α-synuclein-induced translocation, suggesting stimulus-dependent anti-inflammatory mechanisms [50,111].

In the NF-κB pathway, NF-κB is a crucial transcription factor in the inflammatory response that also regulates genes involved in cell proliferation, survival and angiogenesis [104,107]. HTX has been shown to inhibit its nuclear translocation in BV2 microglia cultures, reducing the production of proinflammatory agents such as IL-1β, IL-6, IL-8, IL-22, iNOS, COX-2, TNF-α, NO and PGE2 [9,50,56,86,104,107]. This action contributes to the phenotypic shift of microglia from an inflammatory to a neuroprotective state [56].

In other cell lines, such as J774, RAW 264.7, human endothelial and monocytic cells, HTX prevented NF-κB activation, inhibiting iNOS and COX-2 expression and also reducing cytokine and chemokine secretion [9,107]. HTX butyrate (HTX-B) also attenuated sleep deprivation-induced NF-κB p65 activation and inhibited LPS/ATP-induced NF-κB activation [109]. NF-κB activation is involved in neuronal inflammation and is linked to one of the binding sites in the promoter region of the HO-1 gene, which is regulated by the redox state [107]. In addition, OLE attenuates microglial activation, mainly by modulating NF-κB [9].

The JAK/STAT pathway, together with MAPK/ERK, is also modulated by HTX. The PI3K/Akt and ERK1/2 pathways are involved in the activation of Nrf2/HO-1 by HTX [50]. HTX modulates cytokine signaling, in particular by inhibiting the SOCS and JAK-STAT pathways in RAW 264.7 cells stimulated with lipopolysaccharide (LPS) [9]. A paracetylated derivative of HTX attenuates the LPS-induced inflammatory response through regulation of the Nrf2/HO-1 and JAK/STAT pathways [50]. In breast cancer cells, HTX and OLE can interfere with fast E2-dependent signaling pathways, such as MAPK pathways [62]. Furthermore, it is suggested that HTHB could act by inhibiting the p-JNK/NF-κB p65 signaling pathway, JNK being a type of MAPK [86].

In recent years, the TREM2 (Triggering Receptor Expressed on Myeloid cells 2) pathway has been identified as a new molecular axis relevant in the modulation of brain inflammation and tumor microenvironment. TREM2 is a receptor mainly expressed on microglia, whose activation regulates the transition of these cells towards a disease-associated phenotype, characterized by phagocytic and anti-inflammatory functions [64]. The interaction of TREM2 with its transmembrane adaptor TYROBP triggers intracellular signaling cascades that inhibit NF-κB activation and proinflammatory cytokine production, while promoting phagocytosis of cellular detritus and resolution of inflammation [64]. The interest in the TREM2 pathway in neuro-oncology lies in its dual role: on the one hand, activation of TREM2 can limit chronic inflammation and associated tissue damage; on the other hand, its modulation can influence the antitumor immune response and the plasticity of the tumor microenvironment. Mutations in TREM2 have been associated with increased risk of neurodegenerative diseases and alterations in the microglial response, underscoring its importance as a therapeutic target [63,64].

Recent studies have shown that hydroxytyrosol is able to modulate microglial activity through the TREM2 pathway. In in vitro models, HTX reduces LPS- and α-synuclein-induced microglial activation, decreasing the production of nitric oxide, TNF-α and IL-1β, and promoting the expression of genes associated with an anti-inflammatory and neuroprotective microglial phenotype [63,64]. These effects are accompanied by inhibition of NF-κB signaling, a reduction in NADPH oxidase activity and modulation of the inflammasome, mechanisms in which the TREM2 pathway plays a central role [63,64].

The ability of HTX to promote the phenotypic shift of microglia from a proinflammatory to a resolving and neuroprotective state could have a relevant impact on the brain tumor microenvironment, where neuroinflammation contributes to tumor progression, therapeutic resistance and local immunosuppression. Thus, modulation of the TREM2 pathway by HTX is emerging as a novel and promising mechanism in the control of central nervous system tumor-associated inflammation, complementing its direct effects on tumor cells [63,64]. These specific anti-inflammatory effects on microglia are particularly relevant in the context of brain tumors, where neuroinflammation contributes to tumor progression and therapeutic resistance. The ability of HTX to promote the microglial phenotypic shift from a proinflammatory to a neuroprotective state suggests an additional mechanism of antitumor action that complements its direct effects on cancer cells [63,64].

Taken together, and given that brain tumors actively interact with their microenvironment, the ability of HTX and other polyphenols such as OLE to modulate neuroinflammation and microglial activation could influence the growth and progression of brain tumors.

### 3.3. Induction of Apoptosis and Antiproliferative Effects

Its antioxidant properties and the ability to modulate oxidative stress pathways give HTX a high potential to inhibit tumor growth, but also to induce apoptosis and inhibit cell proliferation in several tumor cell lines, often showing selectivity on non-tumor cells [50]. In vitro studies have revealed that HTX can inhibit the growth of colon carcinoma, breast adenocarcinoma, human leukemia and pancreatic carcinoma cells [50]. HTX-rich extracts have shown antiproliferative and pro-apoptotic effects in the human cancer cell line HL60, avoiding toxicity in non-tumor cell lines. More specifically, HTX has induced apoptotic and antiproliferative activity in human neuroblastoma SH-SY5Y cells, whereas no induction of apoptosis was observed in a normal breast cell line (MCF10A) [50]. HTX derivatives, such as hydroxytyrosyl oleate, have also shown antiproliferative and apoptotic effects in SH-SY5Y cells [50].

Also at the in vitro level, HTX significantly reduces the viability of T98G and U87 glioblastoma cells by inducing apoptosis and cell cycle inhibition [112]. Studies in 3D (tumor spheres) have confirmed that HTX attenuates the phenotype of CD133^+^ glioma stem cells, limiting self-renewal capacity and cell migration [112]. Glioma stem cells (GSCs) represent a critical cell subpopulation responsible for therapeutic resistance and tumor recurrence [113]. Although CD133 has traditionally been considered a marker of GSCs, recent studies reveal greater complexity in the characterization of these cells [114,115]. Single-cell sequencing analysis has identified four main cell states in gliomas: oligodendroglial progenitor cell-like (OPC-like), neural progenitor cell-like (NPC-like), astrocyte-like (AC-like), and mesenchymal (MES-like) [116]. Emerging biomarkers such as GPRC5A show significant association with mesenchymal and stemness features, correlating with poor survival in glioblastoma [117]. Differential expression of CD133 varies according to molecular subtype: the mesenchymal subtype shows low expression of CD133 but high expression of CD44, YKL40, BMI1 and TWIST1, whereas the proneural subtype is characterized by high expression of CD133, OLIG2, SOX2 and EZH2 [116]. The effects of HTX on CD133+ cells observed in preclinical studies suggest a specific therapeutic potential that requires validation in models that consider this molecular heterogeneity [112]. Thus, HTX treatment alone has been observed to decrease cell viability, reducing stem cell ratio and migration in glioblastoma models, albeit less effectively than total olive leaf extract [112].

The derivative HTX-B has effectively inhibited apoptosis induced by the neurotoxin 6-OHDA in SH-SY5Y neuronal cells. This anti-apoptotic effect has been linked to the activation of the Nrf2/HO-1 pathway [9,50,108].

Nrf2 activation contributes to the neuroprotective and anti-apoptotic effects of HTX, and HTX-B inhibits apoptosis through this pathway [9,50,108].

In an experimental mouse skin flap model, HTX reduced tissue apoptosis, evidenced by decreased levels of caspase-3, BAX and cytochrome c [52], promoting a beneficial effect on flap survival. It should not be forgotten that apoptosis is strongly influenced by oxidative stress, and proteins such as BAX, Bak, Bcl-2, Bcl-XL, caspase-3 and cytochrome c play key roles in this process [52,118]. In fact, HTX has been described to promote the intrinsic apoptosis pathway by increasing the expression of the pro-apoptotic proteins p53 and BAX and activating caspases, while reducing anti-apoptotic proteins such as Bcl-2 in tumor cells [112]. In addition, HTX increases apoptosis in the T98G line more than standard temozolomide treatment, and although in some A172 cells its effect was less, overall, it favors programmed death [112].

In addition, the combination of HTX derivatives, such as HTX-ac and HBET, inhibited the heat stress-activated mitochondrial apoptosis pathway by regulating the expression of Bcl-2 family proteins and cytochrome c release, probably through modulation of the PKA-CREB-BDNF pathway [118,119]. HTX also reduced the rate of apoptosis in IL-1β-treated BV2 microglial cells [53]. In a rat model of induced mammary tumors, HTX inhibited tumor growth, which was associated with decreased cell proliferation, as demonstrated by decreased nuclear immunostaining of Ki-67 [44].

On the other hand, although not directly for HTX, OLE has been shown to be a potent inhibitor of cell proliferation in neuroblastoma [50] and breast cancer cells, delaying the cell cycle in the S phase and negatively regulating NF-κB and cyclin D1; HTX and OLE polyphenols can interfere with E2-dependent cell proliferation in breast cancer cells [120] (Figure 4).

### 3.4. Autophagy

Autophagy is an essential process for cellular and tissue homeostasis, responsible for the degradation and recycling of damaged cellular components [52]. Oxidative stress can modulate autophagy by influencing the cell’s ability to remove damaged material. HTX promotes autophagy in different models, such as skin flaps, contributing to its protective and regenerative effects. This promotion of autophagy is mediated by activation of the SIRT1-AMPK-mTOR pathway [52]. In addition, HTX can induce autophagy to alleviate lipid accumulation in the liver [52].

Mitochondria play a critical role in cell function and are involved in both oxidative stress and apoptosis [52,53]. HTX protects mitochondrial function and structure by inhibiting stress-induced release of mitochondrial DNA (mtDNA) to the cytosol [50,53,109]. The release of mtDNA can occur through Bak/Bax channels and activate inflammatory responses such as the NF-κB pathway [109].

HTX also activates sirtuin 1 (SIRT1), a crucial regulator of autophagy, metabolism and aging [52,107]. In skin flaps, HTX stimulated the SIRT1-AMPK-mTOR pathway, increasing SIRT1 levels and the p-AMPK/AMPK ratio, and decreasing the p-mTOR/mTOR ratio, which promoted autophagy; blockade of SIRT1 reversed these effects [52]. HTX can also regulate autophagy through SIRT1-mediated inhibition of Akt/mTOR [52]. The PI3K/Akt pathway is involved in Nrf2/HO-1 activation by HTX [50] and in BDNF-promoted CREB phosphorylation [53,121]. Both PI3K/Akt and AMPK pathways are key mechanisms in the progression of neurodegenerative diseases and represent potential therapeutic targets [9]. A role of AMPK activation has been suggested in the effects of HTX on mitochondrial function and reduction in oxidative stress in the brain of db/db mice [54].

### 3.5. Stress Response

The heat shock factor 1 (HSF1) pathway is a fundamental cellular defense mechanism, coordinating adaptive responses to various types of stress [54,84,96,107]. Its activation, triggered by redox alterations and reactive oxygen species (ROS), is intertwined with the Nrf2 pathway to regulate cytoprotective genes such as heme oxygenase-1 (HO-1) and glutathione-dependent enzymes [107,122]. Compounds such as HTX exert protective effects by inducing mild oxidative stress that activates these pathways without causing significant damage. Beyond oxidative stress, HSF1 acts as a master regulator of the heat shock response, controlling the expression of molecular chaperones such as Hsp70 that maintain proteostasis [54,107]. The involvement of HSF1 also extends to the modulation of neuroinflammation [107]. While curcumin exerts anti-inflammatory effects by negatively regulating the expression of Hsp60 through HSF1, HTX has been shown to suppress NF-κB activation and reduce the production of proinflammatory cytokines such as IL-1β and TNF-α in microglial cells. At the mitochondrial level [107,122], HSF1 participates in the mitochondrial stress response, where HT has shown protective effects by preserving mitochondrial structural integrity, maintaining membrane potential, and preventing inflammation-induced fragmentation. In addition, derivatives such as HTXHB have been shown to be effective in preventing mitochondrial oxidative stress and mitochondrial DNA (mtDNA) release associated with sleep deprivation, highlighting the role of these pathways in maintaining mitochondrial function [107,122]. Together, these findings position HSF1 as a central node in the cellular response to stress, integrating oxidative, proteotoxic, inflammatory, and mitochondrial stress signals. The ability of compounds such as HTX to modulate these pathways opens up promising therapeutic prospects in the management of diseases associated with cellular stress [107,122].

### 3.6. Epigenetic Effects

Epigenetic modulation represents an emerging mechanism in the antitumor action of HTX. There is evidence that this compound can modify the activity of specific regions of cellular DNA through processes such as methylation and histone modification. In particular, it has been observed that HTX, together with derivatives such as HTX-ac and OLE, can induce a significant reduction in the activity of histone demethylase 1 (LSD1), a key enzyme in the removal of methyl groups from histones [123]. Inhibition of LSD1 leads to increased levels of H3K4 mono-methylation, an epigenetic marker associated with gene activation [123]. Although these findings have been described mainly in fibroblasts, modulation of epigenetic enzymes such as LSD1 is a recognized mechanism in the regulation of gene expression and cell behavior, including processes relevant to oncogenesis [123]. Another HTX derivative known as HTX-3 has also shown inhibitory activity on the LSD1 enzyme [123]. On the other hand, the activation of SIRT1 by HTX is also associated with the modulation of epigenetic processes, since SIRT1 is a class III deacetylase [52].

### 3.7. Other Signaling Pathways

Other signaling pathways and genes are also influenced by HTX. In experimental models, it has been associated with increased expression of neuronal plasticity and neurogenesis markers such as BDNF and cAMP response element-binding protein (CREB) [53,107]. It may also influence the expression of enzymes involved in the processing of amyloid precursor protein (APP), relevant in neurodegeneration, illustrating the compound’s ability to modify complex enzymatic pathways [9]. In cancer cells, it has been reported that HTX or OLE can interfere with proliferation and induce apoptosis, modulating pathways such as MAPK, NF-κB and cyclin D1, and possibly interacting with estrogen receptors (ERs) [62].

## 4. Conclusions

HTX is a phenolic compound that has been widely studied for its antioxidant, anti-inflammatory and possible antitumor properties. In the context of CNS tumors, especially glioblastoma, HTX has demonstrated in experimental models the ability to modulate relevant processes such as oxidative stress, inflammation, apoptosis and autophagy, all of which are involved in tumor progression. A key feature of HTX is its ability to cross the BBB, albeit in limited proportions. However, these concentrations may be sufficient to induce relevant biological effects, especially when considering its accumulation in cellular compartments such as mitochondria. In addition, formulation strategies using chemical derivatives or controlled release systems may improve their bioavailability and therapeutic efficacy. Available preclinical data indicate that HTX can act on multiple cell-signaling pathways, affecting both survival and proliferation of tumor cells, with little toxicity on non-transformed cells. This profile suggests that HTX could be useful as an adjuvant to existing therapies, or as part of new combination approaches in brain oncology. Despite abundant preclinical evidence on the effects of HTX in cancer models, clinical evidence remains limited. Existing clinical studies with HTX have focused primarily on cardiovascular and metabolic diseases. While the European Food Safety Authority (EFSA) approved HTX as a novel food ingredient in 2017, establishing its safety for the general population (excluding children under 3 years of age, pregnant and lactating women), the absence of specific clinical trials represents a critical limitation that needs to be addressed by well-designed Phase I/II studies evaluating safety, brain pharmacokinetics and preliminary efficacy in combination with standard therapies.

## Figures and Tables

**Figure 1 cimb-47-00667-f001:**
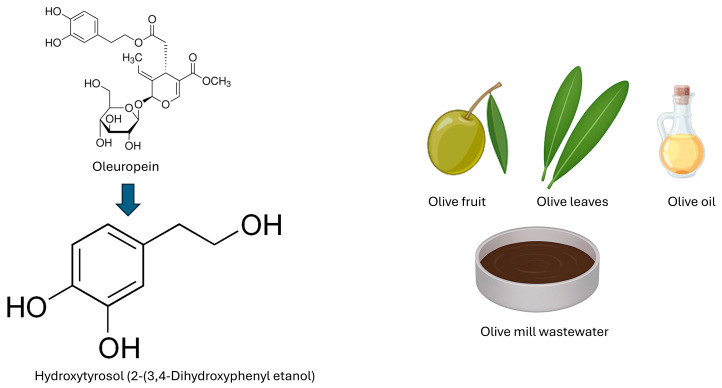
Chemical structure of hydroxytyrosol (2-(3,4-dihydroxyphenyl)ethanol), its derivation from oleuropein, and main natural sources, including olive fruit, olive leaves, olive oil and olive mill wastewater.

**Figure 2 cimb-47-00667-f002:**
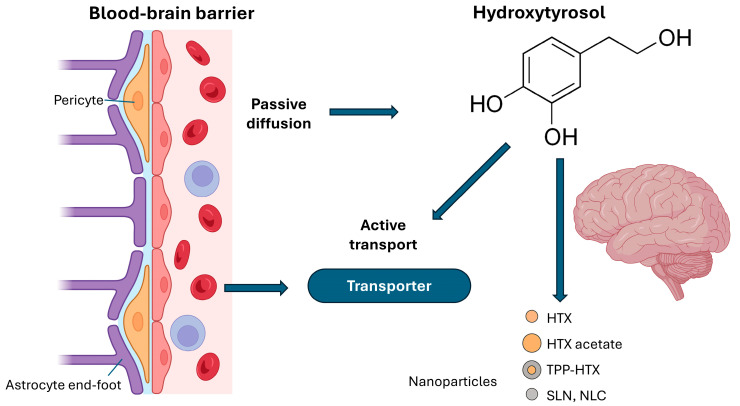
Mechanisms by which hydroxytyrosol (HTX) crosses the blood–brain barrier (BBB). The structure of the BBB includes endothelial cells with tight junctions, pericytes, and astrocytic end-feet. The transport mechanisms of HTX into the brain include passive diffusion, active transport via membrane transporters, and nanoparticle-mediated delivery. Enhanced permeability has been observed with HTX derivatives such as HTX Acetate and TPP-HTX, as well as solid lipid nanoparticles (SLN and nanostructured lipid carriers (NLC)), supporting their potential for targeted brain delivery.

**Figure 3 cimb-47-00667-f003:**
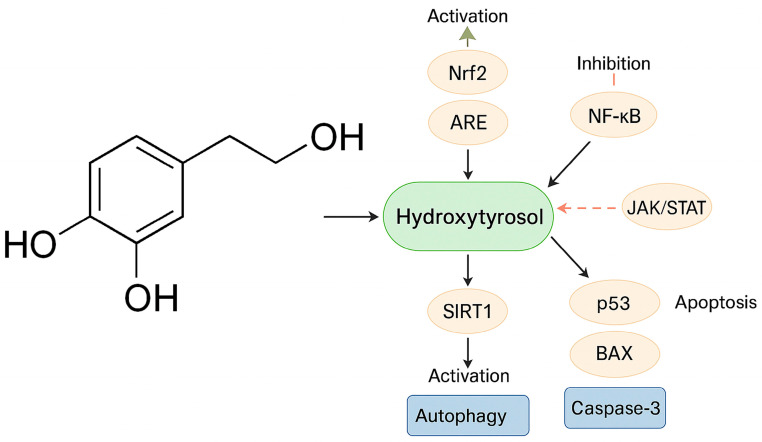
Key signaling pathways modulated by hydroxytyrosol (HTX) in brain tumor models. HTX activates the Nrf2/ARE pathway, promoting antioxidant enzyme expression, and induces autophagy through SIRT1 activation. Simultaneously, it inhibits pro-inflammatory NF-κB and JAK/STAT pathways, reducing cytokine production. HTX also promotes apoptosis by increasing expression of p53, BAX, and caspase-3. This multimodal regulation underlies its antioxidant, anti-inflammatory, pro-apoptotic, and neuroprotective effects.

**Figure 4 cimb-47-00667-f004:**
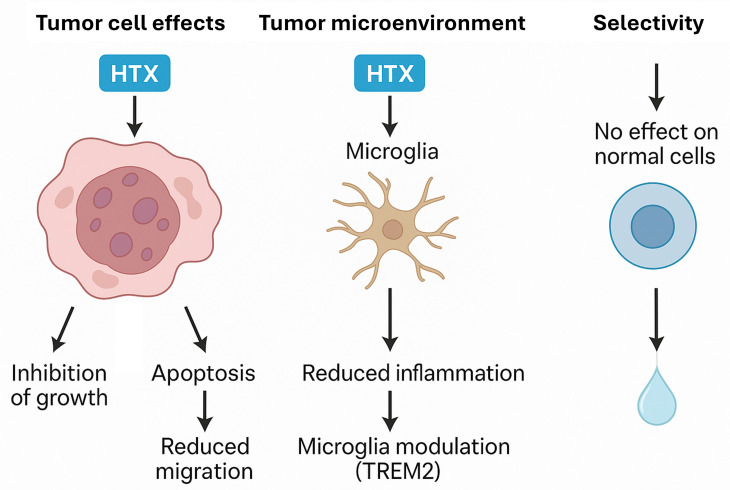
Effects of hydroxytyrosol (HTX) on brain tumor cells, the tumor microenvironment, and normal cells. HTX inhibits tumor cell growth, induces apoptosis, and reduces cell migration. In the tumor microenvironment, HTX modulates microglial activation through the TREM2 pathway, contributing to decreased neuroinflammation. Importantly, HTX exhibits selective activity, sparing normal, non-tumor cells, which highlights its potential as a safe adjunct therapy in brain cancer treatment.

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
