# Peer review of "Hydroxytyrosol and Brain Tumors: Mechanisms of Action and Therapeutic Potential"

_cimb, 2025, doi:10.3390/cimb47080667_

Round 1
Reviewer 1 Report
Comments and Suggestions for Authors
In the manuscript “Hydroxytyrosol and Brain Tumors: Mechanisms of Action and Therapeutic Potential”, the authors describe Hydroxytyrosol (HTX), a popular phenolic olive compound, and possibility of its usage for glioma treatment. The review analyzes in details the situation with conventional therapies in the treatment of brain tumors that have some limitations, and pointed to HTX that could be useful as an adjuvant to existing therapies. The chemical characteristics and mechanisms of action of HTX are described. The text is well written and can be useful for specialists as well as for nonspecialists in this area. The manuscript can be published after correction of some minor grammar errors and typos, such as below.
Minors – missing commas, abbreviation HT instead of HTX…
L13…oxidative stress, inflammation, apoptosis, and autophagy
L15…Hydroxytyrosol (HTX), a phenolic compound present in olives, has shown relevant effects on these processes in experimental models.
L16 …its chemical characteristics, bioavailability, and ability
L20…regulating redox balance, inflammation, programmed cell death, and autophagy.
L80 …to induce damage to DNA, proteins, and cellular lipids.
L290… oral administration of HT - should be HTX
L485… In addition, the combination of HT derivatives, such as HT-ac - should be HTX
L540… In particular, it has been observed that HT, - should be HTX
Author Response
We sincerely appreciate your time and consideration of this paper. The manuscript has been revised to address all typographical errors.
Reviewer 2 Report
Comments and Suggestions for Authors
The authors provide a comprehensive review of the mechanisms of action of hydroxytyrosol and its brain delivery for brain cancer treatment. However, there are some suggestions.
Section 2.1. What is the bioavailability of HTX following oral administration? How many of its metabolites are also active and how does their potency compare with HTX?
Lines 215-216: This expression might be too absolute. Whether an adequate amount of the drug has been delivered into the brain should be decided by comparing the exposure/concentration with the target exposure/concentration. Potency should be taken into account.
Lines 230-232: Please add some quantitative data, such as the brain-to-plasma partition coefficient, to show the extent of brain penetration of the compounds.
Line 237: Please provide the half-life of HTX.
Line 296 subcutaneous glioma: To investigate the treatment of brain tumors, the subcutaneous tumor model has limitations. Are there any studies done in orthotopic brain tumor models?
Author Response
Thank you very much for your helpful suggestions offered as means for improving the paper. We sincerely appreciate your time and consideration of this manuscript.
According to your suggestions, the manuscript has been modified as follow:
- Section 2.1. What is the bioavailability of HTX following oral administration? How many of its metabolites are also active and how does their potency compare with HTX?
Section 2.1 (now section 2.2.) has been modified/completed as follow:
“The bioavailability of HTX is a key factor for its therapeutic potential. Although phenolic compounds usually have a low bioavailability, animal and human studies have shown that HTX is rapidly absorbed in the small intestine and colon, reaching maximum plasma concentrations (Cmax) between 30 and 62 minutes after oral administration. Due to its low molecular weight, it has a large distribution to the brain and other organs [40], where it can exert biological effects in a dose-dependent manner [29,36,40,41]. Additionally, the bioavailability of HTX after oral administration depends on the de-livery matrix. In fact, when administered in olive oil, bioavailability is 99%, while in aqueous solution, bioavailability drops to 75%.
As HTX is rapidly metabolized after ingestion, its conjugated metabolites are the ones mainly detected [36,42,48,49]. HTX undergoes phase II metabolism, mainly through the sulfation and glucuronidation pathways. The sulfation pathway predomi-nates, with exposure between 7 and 9 times higher than that of the original compound, while the glucuronidation pathway has exposure <0.25 times higher compared to the original compound. Only ~11% of the total circulating compounds appear as free HTX. The main conjugated metabolites are two sulfate conjugates and two glucuronide con-jugates. Certainly, these multiple metabolites retain biological activity at the cellular level [29], although with varying potencies compared to the original compound. HTX-3-O-sulfate and HTX-4-O-sulfate maintain significant biological activity (between 60-80% of the original), while HTX-3-O-glucuronide and HTX-4-O-glucuronide retain moderate biological activity (between 40-60% of the original), especially in relation to their antioxidant properties. Some secondary metabolites such as 3,4-dihydroxyphenylacetic acid (DOPAC), homovanillic acid (HVA), and homovanillic alcohol still show antioxidant and cytoprotective activity, although lower than that of the original compound (between 20-40% of the original). Therefore, the metabolism of hydroxytyrosol implies that its biological activity cannot be attributed solely to the original compound, and that it is probably the sulfate conjugates, which represent 73% of the circulating metabolites, that contribute most to the observed benefits [36,42,48,49].
Bioavailability may also….”
- Lines 215-216: This expression might be too absolute. Whether an adequate amount of the drug has been delivered into the brain should be decided by comparing the exposure/concentration with the target exposure/concentration. Potency should be taken into account.
We appreciate the reviewer's comment. The statement in lines 215-216 refers specifically to the results described by Panara et al. (2025) in the cited study. We have clarified this point in the revised manuscript to avoid any possible ambiguity. Thank you for pointing this out.
- Lines 230-232: Please add some quantitative data, such as the brain-to-plasma partition coefficient, to show the extent of brain penetration of the compounds.
We acknowledge the importance of quantitative brain penetration metrics like Kp. However, although it is confirmed that HTX and its metabolites are detected in the brain in vivo, to our knowledge, no studies report brain-to-plasma partition coefficients for these compounds. Available data are limited to qualitative detection of HTX in brain tissue, semi-quantitative measurements without paired plasma ratios or in vitro BBB permeability.
- Line 237: Please provide the half-life of HTX.
HTX half-life has been added as follow:
“….so HTX is excreted mainly by urine and has a short plasma half-life (approximately 2.5 hours and a mean lifetime of around 4 hours) [45]. To overcome….”
- Line 296 subcutaneous glioma: To investigate the treatment of brain tumors, the subcutaneous tumor model has limitations. Are there any studies done in orthotopic brain tumor models?
We thank the reviewer for their constructive observation. Indeed, we recognize that subcutaneous tumor models have inherent limitations for the study of brain tumors, particularly with regard to the blood-brain barrier (BBB), the brain-specific tumor microenvironment, and treatment response, as you rightly point out. Regarding your specific question about existing studies with hydroxytyrosol (HT) or its metabolites in orthotopic brain tumor models, we have conducted an exhaustive literature search in the main databases (PubMed, Scopus, Web of Science). We can confirm that, to date (August 2025), no studies have been published evaluating the effects of HTX or its main metabolites in orthotopic brain tumor models.
Reviewer 3 Report
Comments and Suggestions for Authors
In this review, the authors summarize the characteristics of hydroxytyrosol and its potential in treating brain tumors. Hydroxytyrosol is a phytochemical that commonly found in mediterranean diet, especially in olive or olive -oil contained diet. Increasing evidence has suggested its therapeutic potential as antioxidant and anti-inflammation. While study on its anti-cancer efficacy is limited, a few pre-clinical studies has suggested it may hold potential as alternative in treating intracranial malignancies. This study has well-written. I have the following comments.
- Please include a section for the synthesis and metabolism of HTX. The following link is a good reference for it. https://ift.onlinelibrary.wiley.com/doi/10.1111/1750-3841.14198
- Line 37-76: The paragraph focuses on glioma or glial -derived brain tumor. Please include data for other brain cancers (e.g medulloblastoma) as well.
- Line 529-535: Please expand this section. Please clarify which stress response referred to in this section (e.g mitochondrial stress? ER stress? Unfolded Protein response?)
Author Response
Thank you very much for your helpful suggestions offered as means for improving the paper. We sincerely appreciate your time and consideration of this manuscript.
According to your suggestions, the manuscript has been modified as follow:
- Please include a section for the synthesis and metabolism of HTX.
According to your suggestion, we have included a section about the synthesis and metabolism of HTX, as follow:
“2.1. Synthesis and metabolism of HTX.
The natural biosynthesis of HTX is closely linked to olive (Olea europaea L.), since, although it is present in the plant, most of its content in olives and derivatives comes from the hydrolysis of oleuropein (OLE), a secoiridoid especially abundant in leaves, which contains a structural subunit known as 4-(2-dihydroxyethyl)benzene-1,2-diol, corresponding to HTX [44,48,51,57,58]. OLE is hydrolyzed by endogenous enzymes such as β-glucosidases, releasing glucose, oleuropein aglycone, elenolic acid, and HTX [59]. During olive ripening and oil extrac-tion processes [60], both virgin (VOO) and extra-virgin (EVOO), OLE undergoes several transformations to originate HTX [46,53,57,61]. Their content in EVOO varies between 3.0 and 25.6 mg/kg, depending on factors such as olive variety, climate, fruit maturity, harvest and production process [51,57,62]. Refining removes these compounds, so they are found almost exclusively in VOO and especially in EVOO [44,63]. Olive leaves and by-products from olive processing, such as olive pomace and vegetation water, are al-ternative sources of HTX (figure 1), and methods exist for their large-scale recovery, such as Hidrox® extract obtained by mild acid hydrolysis [55,57].
In addition, the acid hydrolysis of OLE that occurs in the stomach after in-gestion releases glucose and an aglycone that can derive HTX [44,64]. In humans, HTX has been reported to constitute between 96% and 99% of the phenolic metabolites of OLE detected in plasma, underscoring its role as a major source of HTX [48]. In addition, intestinal microflora can degrade OLE in the colon using enzymes such as β-glucosidases, further contributing to the generation of HTX, which can also exert bi-ological effects locally [65].
On the other hand, HTX can be produced endogenously as a byproduct of the ox-idative metabolism of dopamine. Monoamine oxidase (MAO) catalyzes the formation of DOPAL (3,4-dihydroxyphenylacetaldehyde), a neurotoxic metabolite that can be re-duced to HTX by the action of enzymes such as aldehyde reductase. This pathway could represent a physiological detoxification mechanism, as HTX can inhibit both enzymatic and spontaneous oxidation of endogenous dopamine [66].
HTX undergoes extensive hepatic and intestinal metabolism, resulting in multiple conjugated metabolites, which are widely distributed in tissues such as the liver, kidney, lungs, heart, skin, and spleen [67]. Studies have shown that they can accumulate in mitochondria and cross the BBB in small amounts. Some metabolites, such as caffeic acid sulfate, reach relatively high concentrations in the brain, suggesting the existence of selective transport mechanisms [68,69].
The biotransformation of HTX occurs in two main phases. In phase I, hydrolysis and/or hydroxylation reactions occur, catalyzed by enzymes such as cytochrome P450. Hydroxylated and hydrated derivatives can be generated in the liver and intestine. In phase II, conjugation reactions predominate, facilitating solubility and excretion. Sev-eral processes occur [67,69,70]. On the one hand, sulfation, catalyzed by sulfotransfer-ases (SULT), predominates at high doses of HTX. Sulfated conjugates account for ap-proximately 73% of all circulating metabolites. Secondly, glucuronidation is mediated by UDP-glucuronosyltransferases (UGT), which are dominant at low doses. Glucuronides account for only 2-4% of total metabolites, with AUC ratios of 0.15-0.25 relative to HTX. Thirdly, methylation occurs by catechol-O-methyltransferases (COMT), which produces metabolites such as homovanillic acid (HVA) and 3,4-dihydroxyphenylacetic acid (DOPAC). These metabolites reach maximum plasma concentrations approximately 30 minutes after ingestion. Finally, acetylation takes place via acetyl transferases (ACT), which generate HTX acetate (HTX-ac). Among the metabolites identified are 3,4-dihydroxyphenylacetic acid, 3,4-dihydroxyphenylacetaldehyde sulfate, homovanil-lyl alcohol, homovanillic acid, and various glucuronide conjugates [67,69,70].”
- Line 37-76: The paragraph focuses on glioma or glial -derived brain tumor. Please include data for other brain cancers (e.g medulloblastoma) as well.
According to your suggestion, we have included new data about non-glial brain tumors, as follow:
“While gliomas represent the largest group of primary brain tumors, other types of central nervous system neoplasms have distinctive characteristics in terms of incidence, biological behavior, and prognosis. The 2021 WHO classification has introduced signif-icant changes in the categorization of these tumors, establishing specific molecular criteria for several of these neoplastic types [1,29]. Thus, Medulloblastoma is the most common embryonic tumor in childhood, accounting for approximately 20% of all pedi-atric brain tumors. Four main molecular subgroups are distinguished: WNT-activated medulloblastoma, SHH-activated medulloblastoma with wild-type TP53, SHH-activated medulloblastoma with mutated TP53, and non-WNT/non-SHH medul-loblastoma [30]. Each subgroup has distinctive clinical-pathological characteristics and different prognoses. WNT-activated medulloblastoma is associated with a favorable prognosis, with 5-year survival rates exceeding 90%. In contrast, non-WNT/non-SHH tumors show a greater propensity for metastatic spread, with metastatic disease rates at diagnosis of 30-45% and 5-year survival rates of 50-75%. The overall 5-year survival rate for medulloblastoma is approximately 70-80% when the disease has not spread, falling to 60% in cases of metastatic disease. Multimodal treatment includes maximum safe surgical resection, chemotherapy, and craniospinal radiotherapy, with the extent of surgical resection being a significant prognostic factor [1,30,31].
Ependymomas have undergone substantial reclassification in the 2021 WHO clas-sification, with the definition of eight specific types based on anatomical location and molecular characteristics [1,29]. This new classification includes supratentorial ZFTA-fusion-positive ependymoma, supratentorial YAP1-fusion-positive ependy-moma, posterior fossa ependymomas groups PFA and PFB, spinal ependymoma, and MYCN-amplified spinal ependymoma. Survival rates vary significantly depending on molecular subtype and location, with an overall 5-year survival rate of 85%. Spinal ep-endymomas have the best prognosis with a 10-year survival rate of 100%, while su-pratentorial ependymomas have a 10-year survival rate of 20%. In the pediatric popu-lation, 10-year survival is approximately 50-70%. The main treatment includes surgical resection followed by radiotherapy, with chemotherapy reserved for specific cases such as very young children or recurrent disease [1,29,32,33].
Craniopharyngiomas have been reclassified in the WHO 2021 as two distinct enti-ties: adamantinomatous craniopharyngioma and papillary craniopharyngioma, re-flecting their different clinical-demographic and molecular characteristics [1,29]. De-spite being histologically benign tumors (WHO grade 1), they have a high propensity for recurrence and significant morbidity due to their sellar/suprasellar location. The 3-, 5-, and 10-year survival rates are 89.1%, 86.2%, and 83%, respectively. In the pediatric population, the 5-, 10-, and 15-year survival rates are 100%, 98.3%, and 94.6%, respec-tively. Factors associated with better survival include younger age, smaller tumor size, and subtotal resection combined with radiotherapy. Current management tends toward conservative strategies combining limited surgery with adjuvant radiotherapy to minimize hypothalamic morbidity [34-37].
CNS embryonal tumors other than medulloblastoma include atypical tera-toid/rhabdoid tumor (ATRT), embryonal tumor with multilayer rosettes (ETMR), and CNS neuroblastoma with FOXR2 activation [1]. ATRT, characterized by the loss of SMARCB1, has a very poor prognosis with 5-year survival rates of 47.8% in children, 41.5% in adolescents, and 24.6% in adults. ETMR tumors show an overall 5-year survival rate of approximately 25% in retrospective series, reaching up to 66% in patients treated with intensified protocols. Survival is particularly poor in infants and patients with metastatic disease [38].
Diffuse midline gliomas (formerly DIPG) represent highly aggressive brain stem tumors with an extremely poor prognosis. Median survival is 8-11 months, with 2-year survival rates of approximately 10% and 5-year survival rates of 2%. Factors associated with prolonged survival include age younger than 3 years or older than 10 years, shorter duration of symptoms, smaller tumor size, and presence of HIST1H3B mutation [39-41].
Primary CNS lymphomas are predominantly diffuse large B-cell lymphomas with an overall survival of 12-18 months without treatment. With high-dose methotrex-ate-based therapy, 2-year survival rates reach 42-80.8%. However, the prognosis for re-fractory disease or relapse is very poor, with a median survival of 2 months without treatment [42,43].
There is therefore considerable biological and clinical heterogeneity among non-glial brain tumors.”
- Line 529-535: Please expand this section. Please clarify which stress response referred to in this section (e.g mitochondrial stress? ER stress? Unfolded Protein response?)
Thank you for your question, which allows us to better clarify the scope of the “stress response” referred to in this section. It presents the stress response in a broad sense, encompassing the cell's ability to counteract stressful challenges and restore redox homeostasis. For a better explanation, the section has been expanded, as per your suggestion, as follows:
The heat shock factor 1 (HSF1) pathway is a fundamental cellular defense mecha-nism, coordinating adaptive responses to various types of stress [55,85,97,108]. Its ac-tivation, triggered by redox alterations and reactive oxygen species (ROS), is inter-twined with the Nrf2 pathway to regulate cytoprotective genes such as heme oxygen-ase-1 (HO-1) and glutathione-dependent enzymes [108,121]. Compounds such as HTX exert protective effects by inducing mild oxidative stress that activates these pathways without causing significant damage. Beyond oxidative stress, HSF1 acts as a master regulator of the heat shock response, controlling the expression of molecular chaper-ones such as Hsp70 that maintain proteostasis [55,108]. The involvement of HSF1 also extends to the modulation of neuroinflammation [108]. While curcumin exerts an-ti-inflammatory effects by negatively regulating the expression of Hsp60 through HSF1, HTX has been shown to suppress NF-κB activation and reduce the production of pro-inflammatory cytokines such as IL-1β and TNF-α in microglial cells. At the mitochon-drial level [108,121], HSF1 participates in the mitochondrial stress response, where HT has shown protective effects by preserving mitochondrial structural integrity, main-taining membrane potential, and preventing inflammation-induced fragmentation. In addition, derivatives such as HTXHB have been shown to be effective in preventing mitochondrial oxidative stress and mitochondrial DNA (mtDNA) release associated with sleep deprivation, highlighting the role of these pathways in maintaining mito-chondrial function [108,121]. Together, these findings position HSF1 as a central node in the cellular response to stress, integrating oxidative, proteotoxic, inflammatory, and mitochondrial stress signals. The ability of compounds such as HTX to modulate these pathways opens up promising therapeutic prospects in the management of diseases associated with cellular stress [108,121].